# Meta-Learning Online Adaptation of Language Models

**Nathan Hu\***

**Eric Mitchell\***

**Christopher D. Manning**

**Chelsea Finn**

Stanford University
{zixia314,em7}@stanford.edu

## Abstract

Large language models encode impressively broad world knowledge in their parameters. However, the knowledge in static language models falls out of date, limiting the model's effective "shelf life." While online fine-tuning can reduce this degradation, we find that naively fine-tuning on a stream of documents leads to a low level of information uptake. We hypothesize that online fine-tuning does not sufficiently attend to important information. That is, the gradient signal from important tokens representing factual information is drowned out by the gradient from inherently noisy tokens, suggesting that a dynamic, context-aware learning rate may be beneficial. We therefore propose *learning* which tokens to upweight. We meta-train a small, autoregressive model to reweight the language modeling loss for each token during online fine-tuning, with the objective of maximizing the out-of-date base question-answering model's ability to answer questions about a document after a single weighted gradient step. We call this approach **C**ontext-**a**ware **Me**ta-learned **L**oss **S**caling (CaMeLS). Across three different distributions of documents, our experiments find that CaMeLS provides substantially improved information uptake on streams of thousands of documents compared with standard fine-tuning and baseline heuristics for reweighting token losses.

## 1 Introduction

Large language models learn impressively broad world knowledge through large-scale unsupervised pre-training, which they can leverage for a wide variety of downstream tasks (Brown et al., 2020; Chowdhery et al., 2022; Bubeck et al., 2023). However, large language models are typically static artifacts, and as the world changes, the knowledge encoded in their parameters becomes stale. While retrieval-augmented models are one approach to

---
\* Equal contribution.

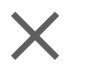

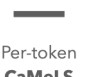

Figure 1: The proposed method CaMeLS learns to rescale the per-token online loss, sparsifying the fine-tuning gradients to emphasize informative timesteps. The middle row shows the weights output by CaMeLS. The **top** and **bottom** rows show raw and weighted per-token gradient norms, respectively.

mitigating the staleness issue, even very large language models often fail to correctly update their memorized predictions when presented with counterfactual retrieved information (Longpre et al., 2021; Li et al., 2022; Si et al., 2023). Moreover, purely parametric language models are uniquely suited for edge computing due to their compact size (relative to a large retrieval index) and simplicity of inference (Gerganov, 2023). Recent work has thus considered variants of online fine-tuning on a stream of documents to efficiently perform direct updates to the knowledge inside of a large language model (Lazaridou et al., 2021; Jang et al., 2022).

Ideally, we could simply fine-tune a language model on an online stream of documents, and the information contained in those documents would be readily available for the model to use in a variety of downstream tasks such as answering questions about the information in the documents. Unfortunately, we find that in this online adaptation setting, fine-tuning with a well-tuned learning rate leads to a nearly negligible improvement in a question-

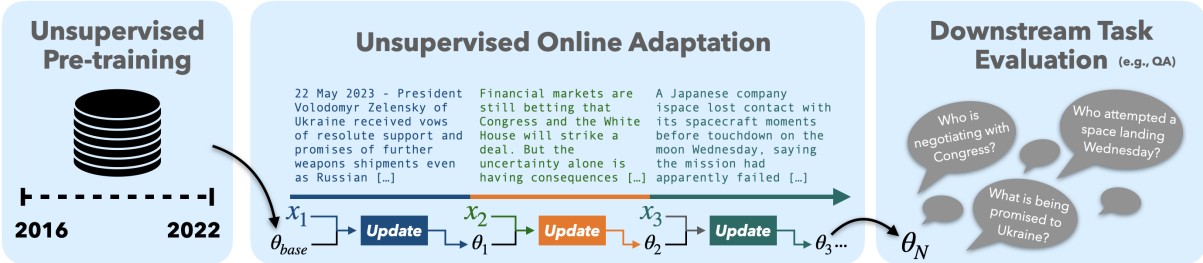

Figure 2: We study the setting of a language model being adapted unsupervised (without annotation of important tokens) on an online stream of documents and being later evaluated on queries (e.g., questions) about those documents. Downstream inputs are **not** provided during the adaptation phase, requiring the model to integrate as much information as possible about the documents.

answering model's ability to answer questions relating to the stream of documents. We hypothesize that naive fine-tuning is not effective in the online adaptation setting because the negative log likelihood (NLL) loss does not accurately reflect the importance of a token. That is, tokens containing important factual information may receive relatively small NLL loss and therefore a small fine-tuning gradient. For example, consider the NLL of the word *Rishi* and the word *Reports* in the phrase *The UK Prime Minister is Rishi Sunak. Reports suggest . . .* for a slightly out-of-date language model. Because Rishi Sunak was a well-known politician before becoming Prime Minister, a model may place reasonably high probability mass on his name (even if other completions are higher probability). On the other hand, 'Reports' will invariably receive low probability, because the distribution over the first word in a sentence is unavoidably high entropy.

This hypothesis suggests that we can improve upon online adaptation by only fine tuning on a subset of tokens which are most likely to lead to useful updates. One natural approach to identify such factual tokens is through salient spans (Guu et al., 2020). Another common technique used to weight words it via TF-IDF scores (Salton and McGill, 1986). We find that fine-tuning while using these heuristics does improve information uptake. However, it is unclear if such heuristic choices are optimal. As an alternative, we explore a method for *learning* a per-token importance weights corresponding to the utility of fine-tuning on that token. However, such utility is difficult to define, and even with a suitable definition, dense per-token annotations of utility are extremely time-consuming to collect. We thus select a definition of utility that enables using *distant supervision* of the utility of each token: a high utility token is one whose fine-tuning gradient improves a question-answering model's

ability to answer questions about the contents of the surrounding document.

Using this notion of a token's utility for online learning, we propose **C**ontext-**a**ware **Me**ta-learned **L**oss **S**caling (CaMeLS), an approach to online adaptation that meta-trains an importance weighting model to identify such tokens in a document. Given a dataset of documents and queries, we use a meta-learning loss to train our weighting model: first, in an 'inner loop,' we update a base model (a proxy for the model we will update at test time) using the gradient of NLL of the document, weighted by the outputs of the importance weighting model. Next, in the 'outer loop', the loss is computed by evaluating the updated base model's performance on the corresponding query. This outer loss is used to updated the parameters of the importance weighting model. During online fine-tuning on a stream of documents, we simply re-weight the online loss using the importance-weighting model's output.

Although the process used to train CaMeLS uses a proxy model (i.e., a stand-in for the model we will update at test time), one might hope that the importance of tokens would be independent of the model used for inner loop updates; to a significant degree, we intuit that the importance of a token should be an innate trait of underlying text. Indeed, we find that the meta-learned importance weights *generalize across models*; for each dataset, we meta-train our importance weighting model once using DistilGPT-2 (Sanh et al., 2019) as the base model and successfully use these weighting model without modification to update GPT-J 6B (Wang and Komatsuzaki, 2021). Across three online adaptation benchmarks based on streams of news and Wikipedia articles, CaMeLS substantially improves knowledge acquisition over naive fine-tuning as well as salient span and TF-IDF based baselines.

## 2 Related Work

Adapting to new data or task distributions is typically studied in the context of continual or lifelong learning (Thrun and Mitchell, 1995; Mitchell et al., 2018). Continual learning in deep networks involves the challenge of simultaneously avoiding *catastrophic forgetting* (McCloskey and Cohen, 1989), the process under which a neural network's performance on old tasks or data is dramatically degraded by the process of learning new information, while maintaining *plasticity* (Dohare et al., 2022), or the ability to adapt to the latest change in the data distribution, even after many changes have already been experienced. While most work in continual learning considers sequences of supervised data (Kirkpatrick et al., 2016; Lopez-Paz and Ranzato, 2017; Shin et al., 2017; Chaudhry et al., 2019), some work also studies continual few-shot (Ren et al., 2021) or unsupervised learning (Rao et al., 2019; Madaan et al., 2022), which is closer to the setting in this paper. However, these works typically focus on streams of visual data.

Dynamic, or streaming, language models were first considered in the context of n-gram language models, combining a cache of recently-used words to update the predictive probabilities of a tri-gram model (Kuhn, 1988; Jelinek et al., 1991; Osborne et al., 2014). Later work describes online EM-based algorithms for efficiently updating n-gram models (Yogatama et al., 2014). Other studies investigate the evolution of decontextualized word embeddings over as a result of temporal shifts in the use of language (Kulkarni et al., 2015; Hamilton et al., 2016) or the use of vector memories to store recent information when training recurrent neural networks online (Rei, 2015). More recently, several studies have explored methods for updating large neural language models, typically through online fine-tuning on a stream of documents (Lazaridou et al., 2021) with architectural constraints (Jang et al., 2022) or explicit conditioning on time (Dhingra et al., 2022) used as strategies to reduce forgetting of old information. Clark et al. (2022) use meta-learning to reduce the compute requirements of online fine-tuning. However, recent work suggests that while increasing the size of language models may largely mitigate the problem of forgetting old information (Driess et al., 2023), improving the efficiency of *acquisition* of new knowledge is still a challenge, and this problem is therefore the focus of the present

work. Other methods for dynamically updating the knowledge in parametric language models develop specialized techniques, called *model editors*, designed to make targeted edits to individual facts (Sinitsin et al., 2020; Mitchell et al., 2021; Meng et al., 2022) or behaviors (Mitchell et al., 2022). However, model editors assume access to annotations of the tokens or facts that must be updated; in this work, we study the problem of *learning* which tokens in an unlabeled sequence of documents are important.

## 3 Meta-Learning Improved Online Adaptation of Large Language Models

Given an out-of-date language model and a stream of recent documents, we aim to update the model such that it effectively answers typical queries about the documents in the stream. By focusing only on retaining knowledge relevant to the 'typical' queries, we avoid the need to completely memorize the documents, making the problem tractable. We study question-answering (QA) models specifically, as the question-answer format makes assessing a model's knowledge straightforward. In this section, we formalize this problem setting and then describe an approach to this setting, Context-aware Meta-learned Loss Scaling.

### 3.1 Unsupervised Online Adaptation

We consider a setting in which an out-of-date model $f_{\theta_{base}}$ is updated with an online stream[2] of recent documents $D_{\text{test}} = \{x_i\}$, ultimately producing an updated model $f_{\theta'}$. The updated model $f_{\theta'}$ is then evaluated with a set of queries $Q_{\text{test}} = \{q_i\}$ with labels $Y_{\text{test}} = \{y_i\}$, where the the $i$th query is drawn from a distribution of queries relating to $i$th document: $q_i, y_i \sim p(q_i, y_i|x_i)$. For example, $q_i$ may be a question about some information in document $x_i$, and $y_i$ the answer to that question implied by the document. Crucially, when using $D_{\text{test}}$ to update $f_{\theta_{base}}$, we do not have access to $Q_{\text{test}}$. Thus, our methodology for updating $f_{\theta_{base}}$ must be broad rather than query specific. In order to make this problem tractable (i.e., not requiring complete memorization of the document stream), we assume that we have an additional corpus of documents $D_{\text{train}}$ and corresponding query samples $Q_{\text{train}}$ and labels $Y_{\text{train}}$ generated by a similar generative process to $Q_{\text{test}}, Y_{\text{test}}$. This training set enables

---

[2]$D_{\text{test}}$ is typically an online stream of documents, but could be an arbitrary ordering over a static collection of documents.

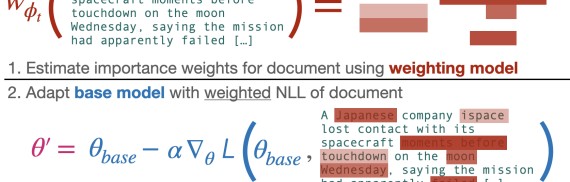

1. Estimate importance weights for document using **weighting model**

2. Adapt **base model** with weighted NLL of document

$$\theta' = \theta_{base} - \alpha \nabla_\theta L \left( \theta_{base}, \text{A Japanese company ispace lost contact with its spacecraft moments before touchdown on the moon Wednesday, saying the mission had apparently failed […]} \right)$$

3. Update **weighting model** to improve **adapted model's** knowledge retention

$$\phi_{t+1} = \phi_t + \eta \nabla_\phi \log p \left( \text{The moon} \middle| \text{Where did ispace try to land Wednesday?}, \theta' \right)$$

Figure 3: A single step of CaMeLS meta-training. In **step 1**, the weighting model (**red**) produces a set of importance weights over the tokens in a given document. In **step 2**, the base model (**blue**) is updated using a single gradient step on the weighted NLL, producing an adapted model (**pink**). In **step 3**, the weighting model is updated to improve the adapted base model's ability to answer questions about the document. During **test-time adaptation**, steps 1 and 2 are applied repeatedly for each document in the test document stream.

learning the *types of queries* that may be of interest, informing how we should update our model to maximize the performance on test queries while minimizing disturbance to its prior knowledge or behaviors. We next describe an algorithm for leveraging this dataset to more efficiently update our base model on the test stream of documents $D_{\text{test}}$.

### 3.2 CaMeLS: Context-aware Meta-learned Loss Scaling

The goal of CaMeLS is to distill the information in the training documents, queries, and labels into a parameter vector $\phi$. This vector summarizes the optimal way to update a base model on a document stream to maximize retention of information *likely to be relevant to test queries*. CaMeLS accomplishes this goal by training a *weighting model* $w_\phi$ (a small autoregressive language model [3]) that re-weights the online NLL loss used in typical online fine-tuning, focusing on the tokens whose NLL gradient is most useful for updating a small proxy base model's knowledge. In other words, the weighting model is trained to re-weight the NLL loss such that the proxy model is able to correctly answer questions about a document after one gradient step on the modified NLL of the document. The weighting model is trained with an episodic bi-level optimization, which we explain next in detail (also see Figure 3).

During each episode, a training document-query-label triple $(x, q, y)$ is sampled from $D_{\text{train}}$ and a

locality example $x_{\text{loc}}$ from $D_{\text{loc}}$. $D_{\text{loc}}$ is a dataset of unlabeled text representing the distribution over which we want the base model's behavior to remain generally unchanged. For all experiments, we use the OpenWebText dataset (Gokaslan et al., 2019) as $D_{\text{loc}}$. Let $\theta_{\text{base}}$ denote the parameters of the proxy base model at the start of the episode. The update to the weighting model involves three steps: 1) computing the weights for the training document, 2) updating the small proxy base model on the weighted NLL on the training document, and 3) backpropagating the 'outer loop' loss[4] of the updated proxy model on a query and label from the training document. These steps are shown in Figure 3. Let $L(f_\theta, x, \mathbf{a})$ denote the weighted NLL of $f_\theta$ on document $x$ using weights $\mathbf{a}$. Steps 1 & 2 are described by the inner loop update rule:

$$\theta' = \theta_{\text{base}} - \alpha \nabla_{\theta_{\text{base}}} L(f_{\theta_{\text{base}}}, x, w_\phi(x)) \quad (1)$$

The inner loop learning rate $\alpha$ can be fixed, sampled, or learned. For all of our experiments, we use a fixed inner learning rate of $\alpha = 5e - 4$. After the updated proxy model is computed, we compute an outer loop loss measuring the effectiveness of the weighted adaptation procedure on the document $x$:

$$L_{\text{outer}} = -\log p_{\theta'}(y|q) + c_{\text{loc}} L_{\text{loc}}(\theta_{\text{base}}, \theta', x_{\text{loc}}) \quad (2)$$

In addition to the negative log likelihood of label given the query and updated base model parameters, the outer loss has a locality term $L_{\text{loc}}$ which prevents the updated base model parameters from excessively changing the base model's behavior. $c_{\text{loc}}$ is set to .1 for all experiments. $L_{\text{loc}}$ is the sum of the KL divergences $L_{\text{loc}}^i$ between the base model before and after adaptation conditioned on each prefix $x_{\text{loc}}^i$ of the locality input $x_{\text{loc}}$, with

$$L_{\text{loc}}^i(\theta_{\text{base}}, \theta', x_{\text{loc}}) = \text{KL}\left(p_{\theta_{\text{base}}}(\cdot|x_{\text{loc}}^i)\|p_{\theta'}(\cdot|x_{\text{loc}}^i)\right) \quad (3)$$

Finally, we perform a single update to the weighting model's parameters by computing the gradient of the outer loop loss with respect to $\phi$. We optimize $\phi$ with the Adam optimizer, using a learning rate of 1e-5. We accumulate outer loop gradients over 24 examples (document-query-label triples) split into 4 batches of 6 triples.

---

[3]Section 3.4 contains details on the weighting model's architecture.

[4]Performing a single update to the proxy model is the inner loop and updating the weighting model according to the updated proxy model's loss on the query is considered the outer loop of a bi-level optimization used to train CaMeLS.

### 3.3 Mitigating Train-Test Shift

The single-step training procedure described above optimizes for effective knowledge retention for a single document. However, in our online adaptation setting, we may update for hundreds or thousands of documents before we evaluate on our downstream queries. In order to mitigate this train-test shift, we modify CaMeLS with two strategies. First, we do not use the same base model parameters during each episode of training. This is done to prevent the weighting model from overfitting to a single base model state. For most training episodes, the starting base model parameters adapted in the inner loop are the final base model parameters in the previous episode. Every $c_{\text{reset}} = 4$ episodes of training, the starting base model parameters are reset to those of the original base model. Second, instead of performing an inner update on a single document, we sample an *inner batch* of $k = 6$ document-query-label triples $(x_1, q_1, y_1), \ldots, (x_k, q_k, y_k)$ for each episode. A sequence of $k$ inner loop updates is performed:

$$\theta_i = \theta_{i-1} - \alpha \nabla_\theta L(f_{\theta_{i-1}}, x_i, w_\phi(x_i)) \quad (4)$$

where $\theta_0 = \theta_{base}$ and $\theta' = \theta_k$. The outer loss is computed as before, but now averaging the query-label loss over the inner batch. By allowing inner loop updates to accumulate during adaptation, $\phi$ learns an updating strategy that preserves the knowledge of prior updates and maintains the base model's ability to learn from subsequent updates.

### 3.4 Compute & Architecture of CaMeLS

Optimizing bi-level objectives like the one used by CaMeLS is rather memory and compute-intensive, requiring memory and compute proportional to the depth of the inner loop (the batch size used for multiple inner loop updates) and proportional to the size of our base/proxy model - each inner loop step creates an updated copy of the base model parameters in the computation graph. However, CaMeLS only requires a lightweight base model; our experiments use DistilGPT-2 as the base model during meta-training, but we find strong transfer to much larger base models during evaluation. The weighting model itself is also small; all experiments use DistilGPT-2 as the weighting model (a MLP with a single hidden state of size 128 is used as the head to produce token weights). Using the base and weighting models described, we are able to train weighting models using 6 inner loop steps on a single NVIDIA A40 GPU.

| Method | Time Per Doc | Total GPU Memory |
|--------|-------------|------------------|
| Uniform | 772.72 ms | 46.62 GB |
| CaMeLS | 782.46 ms | 48.18 GB |

Table 1: Compared to standard uniform fine-tuning, CaMeLS requires slightly more GPU memory to store the weight model and is slightly slower per document. All compute measurements were taken while adapting GPT-2 XL to StreamingQA documents using an 80GB NVIDIA A100.

| Dataset | Avg. text length | Texts per stream |
|---------|-----------------|------------------|
| StreamingQA | ∼510 tokens | 1665 articles |
| SQuAD | ∼150 tokens | 1170 paragraphs |
| ArchivalQA | ∼80 tokens | 3001 paragraphs |

Table 2: Basic statistics of the data in our online document streams. The sample text streams used to evaluate online adaptation vary significantly in length. For the SQuAD and ArchivalQA datasets, the answer to each query is a span in its corresponding document; for StreamingQA, this is not the case.

We next discuss the compute costs of using a trained CaMeLS weighting model for online adaptation. The additional compute needed for CaMeLS is very small compared to uniform fine-tuning — is a single forward pass of a weight model for each document we update on. For large models, the weight model overhead is small compared to the time needed to run a forward and backward pass of the base model. Compared to standard uniform fine-tuning, CaMeLS requires slightly more GPU memory to store the weight model and is slightly slower per document. Table 1 shows compute measurements during online adaptation of GPT-2 XL on StreamingQA.

## 4 Experiments

After outlining datasets and experimental details, we present several experiments aimed at understanding CaMeLS's behavior in unsupervised online adaptation. Section 4.3 studies the extent to which CaMeLS's importance weights improve knowledge retention in online adaptation. Section 4.4 qualitatively and quantitatively explores the weights themselves, suggesting several ablations of CaMeLS that we explore in Section 4.5. Section 4.6 evaluates the cross-dataset generalization of CaMeLS weights, and finally we examine the forgetting and plasticity dynamics of CaMeLS within the document stream in Section 4.7.

### 4.1 Datasets

We apply CaMeLS to three question answering datasets with corresponding source articles. We

partition the datasets into 5 splits. Three of these splits (train, valid, test) are used for training, hyperparameter tuning, and evaluating the CaMeLS weighting model. In order to fine-tune the initial QA base models from generic language models, we reserve two more disjoint splits (in order to prevent reusing questions during initial QA tuning and online adaptation), labeled QA train and QA valid. Additional details on dataset splits and samples are in Appendix A. At evaluation time, a stream of documents is sampled from the test split. The documents length and text stream lengths are shown in Table 2. In the StreamingQA setting, models must adapt to an entire article as opposed to a selected paragraph, making it our most challenging setting.

**StreamingQA** (Liška et al., 2022): The StreamingQA dataset contains a combination of human-written and language model generated questions. Questions are generated from English WMT news articles published between 2007 and 2020.

**SQuAD** (Rajpurkar et al., 2016): The Stanford Question Answering Dataset (SQuAD) contains human generated questions from Wikipedia articles. The answer to each question is a span contained in a paragraph from Wikipedia.

**ArchivalQA** (Wang et al., 2022): The ArchivalQA dataset contains automatically generated questions. Questions are generated from articles in the New York Times Annotated Corpus (Sandhaus, Evan, 2008). The answer to each question is a span contained in an article.

### 4.2 Experimental protocol details

We conducted evaluations on two families of autoregressive language models, the GPT-2 (Radford et al., 2018) and GPT-Neo families (Black et al., 2021), as well as GPT-J (Wang and Komatsuzaki, 2021). We note that all models evaluated use the same text tokenization. For all datasets, we first fine-tune each pretrained model on question-answer pairs from that dataset. These tuned models represent the static language models we wish to update and will be referred to as *base models*. For each dataset, a single weighting model is trained. The proxy language model used during weighting model training is DistilGPT-2 fine-tuned on the QA train split of the respective dataset.

At evaluation time, the base model is updated on a stream of documents sampled from the test split [5]. The final adapted base model is evaluated on

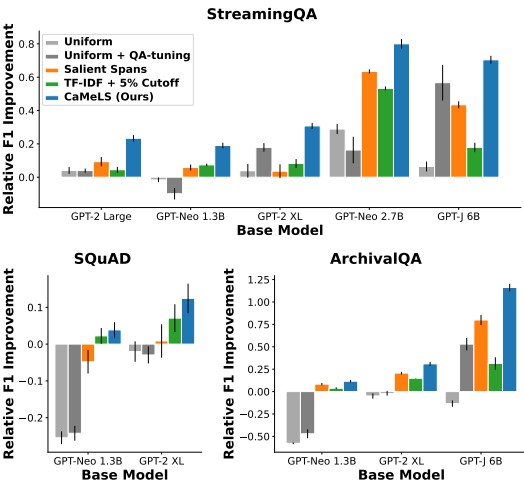

Figure 4: CaMeLS's meta-learned weights improve knowledge uptake after online language model adaptation on a stream of data. The F1 score of the base model before and after adaptation with CaMeLS are computed on questions about the documents used for adaptation. The relative change in F1 is plotted. **Top**, **lower left**, and **lower right** show StreamingQA, SQuAD, and ArchivalQA datasets, respectively. Error bars are standard error over 4 sampled streams of test data.

the questions corresponding to the documents in the sampled stream. We compare CaMeLS with 4 baselines. First is standard fine tuning or **Uniform** where tokens are equally weighted. In **Uniform + QA-tune** we additionally fine tune for question answering after adaptation. Next we consider common weighting heuristics. **Salient Spans** corresponds to assigning a uniform weight to tokens in salient spans and no weight to all other tokens. In **TF-IDF + 5% Cutoff**, we first compute TF-IDF scores using the both the adaptation documents and additional in distribution documents. To account for stopwords, we remove the 5% of words with lowest TF-IDF scores. The remaining TF-IDF scores are used to reweight the tokens. [6] For each combination of base model and online adaptation strategy, the learning rate used at test time was chosen via hyper parameter sweep on a stream of documents sampled from the validation set.[7]

### 4.3 CaMeLS improves knowledge retention

We first compare the knowledge retained by CaMeLS and baselines for three different data distributions in Figure 4. CaMeLS outperforms other

---

[5]We use an Adam Optimizer most experimental runs. Due

to compute constraints, we use an Adafactor optimizer for adaptation of GPT-Neo 2.7B and GPT-J 6B.

[6]TF-IDF scores are computed using a word level tokenization. These scores are then mapped to the BPE tokenization of the adapted language models.

[7]All learning sweeps are conducted over the following values: [1e-4, 2.5e-5, 6.25e-6, 1.625e-6]. The optimal learning rates are in Appendix D.

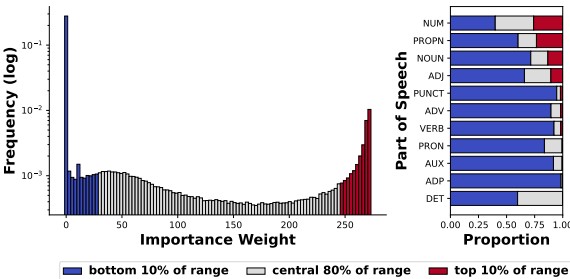

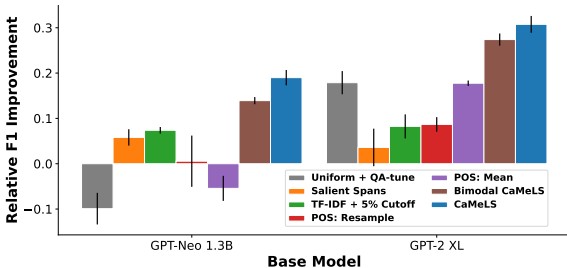

Figure 5: The importance weight distribution learned by CaMeLS is bimodal, with proper nouns and numbers being the parts of speech most likely to have high importance weights. The overall importance weight distribution **(left)** and the distribution conditioned by part of speech **(right)** are shown on the validation split of StreamingQA.

Figure 6: Ablations of CaMeLS. **Bimodal Ablation** restricts the weighting model from outputting intermediate values while the **POS** ablations remove context dependence, conditioning only on part of speech. While restricting CaMeLS to output only one of two values only slightly reduces performance, conditioning only on part of speech, rather than full context, drastically reduces knowledge retention.

online adaptation approaches across a range of datasets and weighting models. Despite the difference in scale between the proxy model used during weight training and the evaluated base models, CaMeLS's learned importance weights generalize well to the largest base model we evaluate, GPT-J 6B, which is over 70 times the size of the proxy model (DistilGPT-2, 82 million parameters) used during training. We find that standard online fine tuning (uniform weighting) with Adam performs very poorly on online adaptation. Even with a tuned learning rate and further training for question answering post adaptation, uniform weighting fails to achieve a significant improvement for several models tested.

### 4.4 Analysis of learned weights

One benefit of CaMeLS over other methods for meta-learning model updating strategies is that learned updating strategy, token weights, is interpretable. Figure 1 shows the per-token weights on sample text and how they combine with the unweighted gradient norms to produce sparsified per-token gradient norms. In this section, we provide additional analysis of CaMeLS's learned weights. We examine the distribution of weighting model outputs on articles in the validation set of StreamingQA in Figure 5. As our qualitative evaluations show, we confirm that the distribution of weights over the entire validation split of StreamingQA is indeed sparse and bimodal. We thus interpret the weighting model as acting as a context-aware binary classifier, determining if a token is informative or uninformative. When binning weights by part of speech, we find that numbers and proper nouns are most frequently assigned a high weight. This result aligns with Lazaridou et al. (2021), who found that

an outdated language model's performance most rapidly declines on proper nouns and numbers.

### 4.5 Ablations

In order to verify that context-aware weights are truly necessary to achieving improved knowledge retention, we now examine several ablations of CaMeLS. In the **POS: Resample** ablation, the weight of each token is generated by sampling from the distribution of importance weights on all tokens of the same part of speech. In the **POS: Mean** ablation, each token is weighted by the mean importance weight assigned to tokens of that part of speech. We additionally consider a **Bimodal** ablation where outputs of the weighting model are rounded to either the largest or smallest value in the distribution of importance weights.

Figure 6 shows the results on the StreamingQA dataset for two different base models. We observe that ablating the weighting model to only output two values slightly reduces performance, while still achieving significant F1 improvement and outperforming baseline approaches. The strong performance of the binary ablation suggests that a binary decision of whether to train on a given token is an effective approach to online adaptation, though the full version of CaMeLS that allows for variation in the weight magnitude still performs best.

In contrast, neither part-of-speech ablation produces effective knowledge retention, either performing worse than the uniform baseline or failing to significantly increase F1 score. This result strongly suggests that although part of speech correlates strongly with learned weights, *part of speech alone is not sufficient to determine when a token contains important information*. We conclude that context-awareness is indeed helpful for identifying

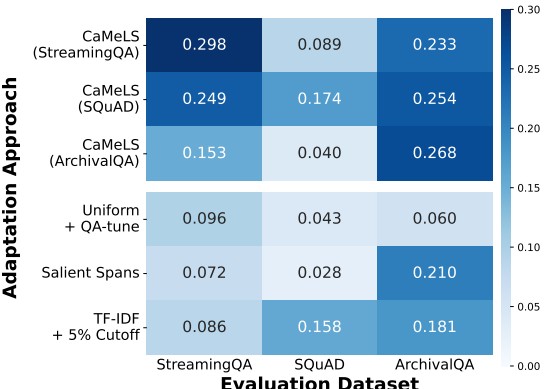

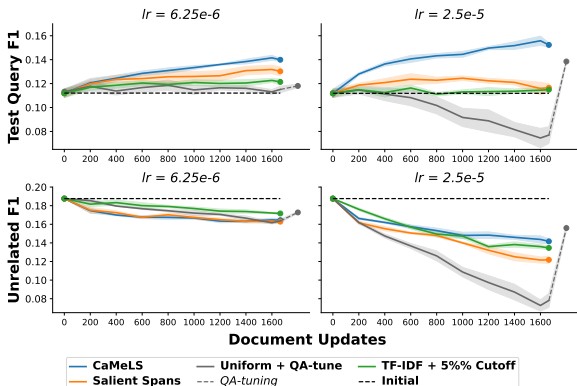

Figure 7: CaMeLS weight models on unseen data distributions (off-diagonals of top three rows) frequently outperforms baseline online adaptation approaches (bottom three rows). Each CaMeLS model was trained on a single dataset (shown in parenthesis) and used to adapt GPT-2 XL on streams of data from various datasets.

important tokens in online adaptation.

## 4.6 Cross Dataset Transfer

Beyond generalizing to new base models, we now study CaMeLS's ability to generalize to new data distributions. We evaluate CaMeLS's performance for all nine possible combinations of train and test dataset, using StreamingQA, SQuAD, and ArchivalQA. Figure 7 shows the results. We find that CaMeLS trained on a *different dataset* still typically outperforms the baseline methods, providing stronger evidence that the weighting scheme learned by CaMeLS is general-purpose. The generalizability of CaMeLS's weighting model is a key attribute increasing its practical utility.

## 4.7 Forgetting and plasticity

So far, our evaluations have considered only the QA accuracy at the end of online adaptation. In this section, we investigate the evolution of learning *during* the online adaptation process. While adapting GPT-2 XL to data from StreamingQA, we evaluate the intermediate models produced by CaMeLS and baseline methods every 200 document updates. Results are plotted for two learning rates. 6.250e-6 is the optimal learning rate for the TF-IDF baseline while 2.500e-5 is the optimal learning rate for all other methods shown. Figure 8 shows the performance when intermediate models are evaluated on the *entire* set of evaluation queries and additionally evaluated on a set of unrelated queries sampled from the QA validation spit. CaMeLS consistently improves performance on test queries during online adaptation, while the best performing baseline —

Figure 8: Base model performance during StreamingQA online adaptation of GPT-2 XL. Performance is evaluated every 200 article updates on the downstream answering task **(top)** and on unrelated validation questions used in QA pretraining **(bottom)**. Results are plotted for two learning rates. 6.250e-6 **(left)** is the optimal learning rate for the TF-IDF baseline while 2.500e-5 **(right)** is the optimal learning rate for all other methods shown. Shaded regions are 1 standard error over 4 runs. All adaptation methods lead to gradual degradation in unrelated questions performance. CaMeLS results in gradual increases in base model test performance. Using its optimal learning rate, uniform fine-tuning with post adaptation QA tuning are only realizes its performance increases after a post-adaptation QA-tuning step.

uniform fine-tuning with a learning rate of 2.500e-5 and additional QA-tuning — results in gradual degradation in test performance with improvement only becoming realized after the post-adaptation QA-tuning step. Turning to performance on unrelated queries, we see that all methods result in a gradual degradation in performance on independent queries. At a learning rate of 6.250e-6, all methods lead to comparable degradation in performance on unrelated queries. At a learning rate of 2.5e-6 CaMeLS leads to the lowest drop in unrelated query performance. Taken together, these results suggest that the CaMeLS is able to more effectively update the base model's knowledge, while still preserving the model's pre-existing knowledge and its representation of the task.

Finally, in Figure 9, we aim to answer the questions *how long does the model remember the answer to a question after observing it?* We show the average improvement in F1 score across test queries against the number of timesteps since the model observed the document containing the answer to the query. Each adaptation method is applied using a uniquely tuned learning rate. After the 200 document sequence containing the relevant document, all methods see a clear average improvement in F1 score, signifying learning is happening. However, we also note that CaMeLS

produces both a higher *initial improvement* as well as a higher *asymptotic improvement* in F1 score. CaMeLS not only improves the immediate plasticity of the model, integrating knowledge more readily, but also reduces forgetting, preserving the newly-integrated knowledge for longer.

# 5 Discussion

While large language models are powerful, keeping them up-to-date remains a challenge. In this paper, we consider the unsupervised online language model adaptation setting, in which a language model's knowledge must be updated using a stream of documents, without annotations of key facts or information. Finding that naive online fine-tuning provides little retention of knowledge from the document stream, we propose Context-aware Meta-learned Loss Scaling (CaMeLS), a meta-learning algorithm that learns an importance weighting model to reweight the per-token loss of the online data stream. CaMeLS leverages side information of the form of paired documents and knowledge queries about those documents to identify which tokens in the documents are most likely to be informative for answering downstream queries. Empirically, we find that the importance weighting model learned by CaMeLS consistently improves knowledge retention across three datasets of documents and questions. Crucially, we find that CaMeLS's importance weighting model *generalizes* across outdated language models and datasets, meaning that an importance weighting model can be trained once on a small proxy language model (such as DistilGPT-2) and then be immediately used to improve online adaptation of much larger models, like GPT-J 6B. This transferrability of CaMeLS's weighting model significantly increases its practical utility.

## Limitations & Future Work

While our experiments suggest that learned importance weights consistently improve knowledge retention after unsupervised online adaptation, our study has several limitations. CaMeLS assumes access to side information in the form of training document, query, and label triples. This requirement may be onerous in domains where labeling is expensive. Future work may apply CaMeLS to settings without access to side information queries and labels, i.e., only a purely unlabeled stream of training documents, using the temporal structure of the data as the signal for learning. We study

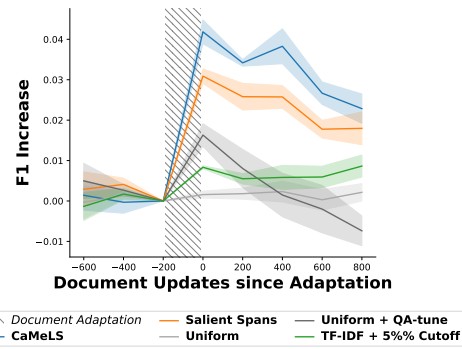

Figure 9: While adapting GPT-2 XL on StreamingQA, we examine the average improvement in F1 score of queries against the time since the model observed the corresponding document. The shaded region represents the interval in which the source document was presented. CaMeLS leads to a larger *initial* improvement and *asymptotic* improvement in F1 score than other methods. Although this mid-adaptation evaluation does not use QA-tuning, Uniform + QA-tune corresponds to uniform fine tuning using a learning rate optimized to downstream performance given an additional QA-tuning step. Each adaptation method is applied using a uniquely tuned learning rate.

adaptation on steams of thousands of documents. However, in order to effectively update outdated language models in real-world scenarios, it is reasonable to expect a significantly larger volume of documents. Beyond dataset scale, our experiments study adaptation of base models up to 6B parameters, but recent work suggests the continual learning dynamics of language models changes drastically at extreme scale (100B+ parameters); future work may increase the scale of the present study by considering adaptation on longer streams of documents using larger base evaluation models. Finally, we study only question-answering models and the question-answering task, as it is the most direct form of knowledge retention assessment. Future work may examine knowledge retention in other types of models through alternative downstream tasks that leverage the knowledge in the document stream more indirectly, as well as studying the ability to continually update general-purpose generative models of language or dialogue models.

## Acknowledgements

The authors thank Huaxiu Yao for his input at multiple stages of the project. CF and CDM are CIFAR Fellows. EM gratefully acknowledges funding from a Knight-Hennessy Graduate Fellowship. This research was supported in part by Juniper Networks.

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

| Split | SQA $N_{x/q}$ | SQuAD $N_x$ | $N_q$ | ArchivalQA $N_x$ | $N_q$ |
|---|---|---|---|---|---|
| Train | 21k | 8.6k | 39.9k | 12.8k | 21.7k |
| Validation | 1.7k | 1.2k | 5.6k | 3.0k | 5.3k |
| Test | 5k | 2.1k | 10.6k | 5.0k | 8.7k |
| QA Train | 40k | - | 40k | - | 12.4k |
| QA Valid. | 4k | - | 2.1k | - | 3k |

Table 3: Number of documents $N_x$ and questions $N_q$ for each dataset. Each document in StreamingQA (SQA) corresponds to a single question, while SQuAD and ArchivalQA contain documents corresponding to multiple questions.

## A  Dataset Details

The sizes of dataset splits are shown in Table 3. Sample documents, questions, and answers are shown in Table 4. Only documents from 2018 and on-wards are used to the train, validation, and test splits of StreamingQA. For SQuAD, the entirety of the validation set of SQuAD is used at our test split. The topics in training set of SQuAD are re-partitioned to form the other 4 splits. We divide the validation set of the ArchivalQA dataset to form our 5 splits. These splits are done temporally, using documents from 1987-1990 for QA Training, 1991-1992 for QA Validation, 1993-2001 for Training, 2002-2003 for Validation, and 2004-2007 for Testing.

## B  Larger Proxy Models

We conduct a preliminary investigation on the effect of using a larger proxy model during CaMeLS meta-training. By default, we use a QA-tuned DistilGPT2 (82M) as the proxy model. We additionally meta-train using a GPT-2 Small (117M) as the proxy model. Due to compute limitations we were not able to meta-train using any larger proxy models. Results on StreamingQA are shown in table 5. We see no significant difference in performance in this setting. Qualitatively, the two weighting models generate similar outputs. We hypothesize that CaMeLS learns a weighting which reflects the innate importance of tokens in the text to answering the meta-training questions, rather than a proxy model specific token importance. We emphasize that this is a hypothesis and believe a more rigorous exploration of proxy model size is an exciting direction for future work.

## C  Combining CaMeLS with other online Adaptation Methods

There are various other methods for online adaptation which leverage the adaptation documents. Two such methods are in-context learning and retrieval. This section shows preliminary experiments lever-

| Dataset | Document | Question | Answer |
|---|---|---|---|
| StreamingQA | Colin Farrell goes missing in new trailer March 2 (UPI) – Colin Farrell joins the cast of Artemis Fowl in the latest trailer for Disney's upcoming fantasy-adventure film. Farrell is featured in the clip, released on Monday, as the missing father of Ferdia Shaw's Artemis Fowl who also goes by the same name. Farrell's character is a criminal mastermind who has mysteriously disappeared. Artemis Fowl learns that his father has protected powerful secrets that have kept mankind safe and learns that his disappearance is connected to a secret fairy world. Artemis Fowl, with help from his loyal protector Butler (Nonso Anozie), embarks on a dangerous journey into the unknown in order to save his father. . . | What does Artemis Fowl embark on? | a dangerous journey into the unknown |
| SQuAD | Luther is honoured on 18 February with a commemoration in the Lutheran Calendar of Saints and in the Episcopal (United States) Calendar of Saints. In the Church of England's Calendar of Saints he is commemorated on 31 October. | When is Luther commemorated in the Lutheran Calendar of Saints? | 18 February |
| ArchivalQA | If it feels like the Heat Miser ("Oh, some like it hot, but I like it really hot") has been lurking of late, it may be due to NBC's coming remake of the animated 1974 television movie "The Year Without a Santa Claus." The four-time Tony Award winner Harvey Fierstein ("Hairspray") signed on this week to replace Chris Elliott in the role of the Heat Miser; Mr. Elliot had to bow out because of a scheduling conflict. The new version will be seen later this year. | Who replaced Chris Elliott as the Heat Miser? | Harvey Fierstein |

Table 4: Example documents, questions, and answers from the test split of each dataset.

| Proxy Model | GPT-Neo 1.3B | GPT-2 XL |
|---|---|---|
| DistilGPT2 (82M) | $0.190 \pm 0.017$ | $0.308 \pm 0.018$ |
| GPT-2 Small (117M) | $0.176 \pm 0.023$ | $0.309 \pm 0.012$ |

Table 5: StreamingQA F1 Increase comparison for CaMeLS meta-trained using DistilGPT2 (82M) and GPT-2 Small (117M) proxy models. Online adaptation of GPT-Neo 1.3B and GPT-2 XL is evaluated. In the tested setting, varying the proxy model size does not change CaMeLS performance.

| Method | GPT-2 XL | GPT-Neo 1.3B |
|---|---|---|
| 5-shot ICL | 0.1091 | 0.0533 |
| ICL w/ CaMeLS | 0.1594 | 0.1398 |

Table 6: Adapting the base models with CaMeLS consistently improves the F1 scores in a simple in-context learning setting.

aging CaMeLS in conjunction with these methods on the ArchivalQA dataset. We show that CaMeLS is complementary to both in-context learning and retrieval; for both methods, the adaptation performance is improved by CaMeLS.

In our first set of experiments, we do five-shot in-context learning. We assume we can prompt the model with the oracle document containing the answer to the question (i.e., the best-case scenario for in-context learning). The prompt is formatted as `[ex. doc 1] [ex. q 1] [ex. ans 1] ... [ex. doc 5] [ex. q 5] [ex. ans 5] [oracle test doc] [test question]`. We use the base GPT-2 XL and GPT-Neo 1.3B models (QA-tuned models performed much worse with in-context learning). As shown in Table 6, we find that adapting the base models with CaMeLS consistently improves the F1 scores of in-context learning.

In a second set of experiments, we consider a simple retrieval setup. Results are shown in Table 7. We fine-tune GPT-2 XL and GPT-Neo 1.3B to answer questions with the source document in the context. We retrieved documents using random, oracle, and BM25 document retrieval. We use CaMeLS to update the parameters of the document-conditioned question-answering models. Across

models and retrievers, Using CaMeLS to adapt document-conditioned question-answering models consistently improves adaptation performance over vanilla retrieval.

These results use the CaMeLS weighting model trained using a QA-proxy model on ArchivalQA. We expect the performance of CaMeLS to increase if meta-trained using a proxy model and outer loss more analogous to the evaluation setting. For example, increase performance in the retrieval setting, we could present the source document when computing the outer loss and using a document conditioned QA proxy model. We acknowledge that we do not evaluate any baseline methods and think that extensive comparisons of parametric updating in conjunction with these other methods would be an exciting direction for future work. As is, these results do show that parametric online adaptation can be used to complement document-storage based methods.

## D Optimal Online Adaptation Learning Rates

When evaluating loss reweighing methods in our experiments, the learning rate used to adapt our base models is found via a learning rate sweep. For each combination of dataset, base model, and adaptation method, we test a range of learning rates to adapt the base model on a stream of documents from the validation split of the dataset. The best performing learning rates are used for later experiments on the test split of the dataset. We test the following learning rates: [1e-4, 2.5e-5, 6.25e-6, 1.625e-6]. The optimal learning rates found via these sweeps are shown in Figure 10.

| Method | GPT-2 XL | | | GPT-Neo 1.3B | | |
|---|---|---|---|---|---|---|
| | Random | BM25 | Oracle | Random | BM25 | Oracle |
| Vanilla Retriever | 0.0694 | 0.6812 | 0.7290 | 0.0624 | 0.6898 | 0.7401 |
| Retriever w/ CaMeLS | 0.1156 | 0.7106 | 0.7565 | 0.1045 | 0.7356 | 0.7832 |

Table 7: Using CaMeLS to adapt document-conditioned question-answering models consistently improves adaptation performance over vanilla retrieval.

### (a) StreamingQA

| Method | Base Model | Learning Rate |
|---|---|---|
| CaMeLS | GPT-2 Large | 2.500e-5 |
| | GPT-Neo 1.3B | 6.250e-6 |
| | GPT-2 XL | 2.500e-5 |
| | GPT-Neo 2.7B | 6.250e-6 |
| | GPT-J 6B | 6.250e-6 |
| TF-IDF + 5% Cutoff | GPT-2 Large | 1.625e-6 |
| | GPT-Neo 1.3B | 1.625e-6 |
| | GPT-2 XL | 6.250e-6 |
| | GPT-Neo 2.7B | 1.625e-6 |
| | GPT-J 6B | 1.625e-6 |
| Salient Spans | GPT-2 Large | 6.250e-6 |
| | GPT-Neo 1.3B | 1.625e-6 |
| | GPT-2 XL | 2.500e-5 |
| | GPT-Neo 2.7B | 1.625e-6 |
| | GPT-J 6B | 6.250e-6 |
| Uniform + QA-tuning | GPT-2 Large | 1.625e-6 |
| | GPT-Neo 1.3B | 2.500e-5 |
| | GPT-2 XL | 2.500e-5 |
| | GPT-Neo 2.7B | 6.250e-6 |
| | GPT-J 6B | 2.500e-5 |
| Uniform | GPT-2 Large | 1.625e-6 |
| | GPT-Neo 1.3B | 1.625e-6 |
| | GPT-2 XL | 1.625e-6 |
| | GPT-Neo 2.7B | 1.625e-6 |
| | GPT-J 6B | 1.625e-6 |

### (b) SQuAD

| Method | Base Model | Learning Rate |
|---|---|---|
| CaMeLS | GPT-Neo 1.3B | 6.250e-6 |
| | GPT-2 XL | 6.250e-6 |
| Salient Spans | GPT-Neo 1.3B | 1.625e-6 |
| | GPT-2 XL | 1.625e-6 |
| TF-IDF + 5% Cutoff | GPT-Neo 1.3B | 1.625e-6 |
| | GPT-2 XL | 1.625e-6 |
| Uniform | GPT-Neo 1.3B | 1.625e-6 |
| | GPT-2 XL | 1.625e-6 |
| Uniform + QA-tuning | GPT-Neo 1.3B | 1.625e-6 |
| | GPT-2 XL | 1.625e-6 |

### (c) ArchivalQA

| Method | Base Model | Learning Rate |
|---|---|---|
| CaMeLS | GPT-Neo 1.3B | 1.625e-6 |
| | GPT-2 XL | 6.250e-6 |
| | GPT-J 6B | 6.250e-6 |
| Salient Spans | GPT-Neo 1.3B | 1.625e-6 |
| | GPT-2 XL | 1.625e-6 |
| | GPT-J 6B | 6.250e-6 |
| TF-IDF + 5% Cutoff | GPT-Neo 1.3B | 1.625e-6 |
| | GPT-2 XL | 1.625e-6 |
| | GPT-J 6B | 6.250e-6 |
| Uniform | GPT-Neo 1.3B | 2.500e-5 |
| | GPT-2 XL | 6.250e-6 |
| | GPT-J 6B | 6.250e-6 |
| Uniform + QA-tuning | GPT-Neo 1.3B | 2.500e-5 |
| | GPT-2 XL | 6.250e-6 |
| | GPT-J 6B | 6.250e-6 |

Figure 10: Optimal adaptation learning rates used to evaluate each combination of adaptation method and base model.