# OpenReview forum: "Meta-Learning Online Adaptation of Language Models"
_EMNLP/2023/Conference — EMNLP 2023 Main_

### Official Review · Reviewer_2YUV · 2023-07-22

**Typos Grammar Style And Presentation Improvements:** 1. More examples such as Figure 1 can…
**Soundness:** 3

**Excitement:**

3: Ambivalent: It has merits (e.g., it reports state-of-the-art results, the idea is nice), but there are key weaknesses (e.g., it describes incremental work), and it can significantly benefit from another round of revision. However, I won't object to accepting it if my co-reviewers champion it.

**Missing References:**

None

**Paper Topic And Main Contributions:**

This paper considers the unsupervised online language model adaptation setting and proposes a meta-learning algorithm called (CaMeLS) that learns an importance weighting model to reweight the per-token loss of the online data stream. The experiments show the effectiveness of the proposed method.

**Questions For The Authors:**

Question A: Since a training set is required in your setting, what are the key differences between Online Adaptation and Transfer Learning?


**Reasons To Accept:**

1. Introducing meta learning for unsupervised online language model adaptation is innotative and reasonable.

2. The proposed method only requires training a much smaller proxy model for token weight estimation. And the quantitative and qualitative results show that the proposed method improves the model ability on new quires given new documents.

**Reasons To Reject:**

1. The setting of Unsupervised Online Adaptation is a little bit strange. As described in Sec 3.1, the model requires a training set, including documents, quires and labels. It seems that the adaptation process is NOT "Unsupervised" because the training set also requires annatations.

2. The problem that this paper focuses on may be unrealistic. Figure 8 shows that adapting to test documents leads to performance degradation on unrelated queries. In practice, we expect to update the knowledge of Large Language Models without affecting the performance on general tasks. Besides, existing large scale QA systems, e.g. GPT-4, show strong In-Context Learning abilities. In other words, the model can reason about new documents and answer questions that have never been seen before.

**Reproducibility:**

3: Could reproduce the results with some difficulty. The settings of parameters are underspecified or subjectively determined; the training/evaluation data are not widely available.

**Reviewer Confidence:**

3: Pretty sure, but there's a chance I missed something. Although I have a good feel for this area in general, I did not carefully check the paper's details, e.g., the math, experimental design, or novelty.

---

> ### Author Rebuttal · Authors · 2023-08-29
>
> Thank you for your review! We're glad you appreciated the innovation and effectiveness of meta-learning for online LM adaptation.
>
> Regarding the question of unsupervised online adaptation, we would like to clarify that while the meta-training procedure utilizes training data with labels (in a relatively common format of document + question + answer), adaptation itself is indeed assumed to be unsupervised, that is, the model is fine-tuned only with unlabeled documents (unlike supervised transfer learning, in which we would typically fine-tune on labeled examples). However, even the meta-training process does not include explicit labels of the *important information in each document*; thus in this sense we might consider even the meta-learning phase as well to be only distantly supervised. Further, our experiments show that the result of this distantly-supervised stage can be re-used for many models, so this stage only needs to be performed once, and we are able to re-use the produced weighting model to update many different LMs in an unsupervised fashion. We will clarify this distinction and nomenclature in the revised version of the paper!
>
> Regarding the realism of the setting, we note that fine-tuning on new data without forgetting has long been considered a fundamental challenge for neural networks, e.g. since McCloskey and Cohen's pioneering work on catastrophic forgetting [1]. Numerous subsequent works have validated the difficulty of continual learning of neural networks without forgetting, e.g. [2], or [3] for a review. While increasing model scale has been shown to reduce catastrophic forgetting [4], even very large language models show non-trivial forgetting: Fig. 6 in [4] shows a reduction in language generation task performance of over 60% even for a 62B parameter model (similar in size to the largest open-source language models) after adapting to a new suite of embodied tasks. Even a 540B parameter model shows a roughly 4% degradation in performance after adaptation.
>
> For these reasons, we do not find it surprising that fine-tuning on new documents leads to degradation of previous knowledge.
>
> Regarding in-context learning, we note that our setting involves unsupervised adaptation to a long sequence of documents (more than one thousand), which is much larger than GPT-4's context window. Even if the model had a longer context window, [5] shows that models with context lengths long enough to condition on this many documents are not generally able to effectively leverage the information in long contexts. Finally, [6] show that gradient-based
> adaptation can provide more robust uptake of knowledge compared with in-context learning, motivating our emphasis on parametric knowledge updating.
>
> Finally, thanks for the useful suggestions on improving the paper presentation.
>
> [1] Catastrophic Interference in Connectionist Networks: The Sequential Learning Problem. Michael McCloskey, Neal J. Cohen, 1989. Psychology of Learning and Motivation.\
> [2] Overcoming catastrophic forgetting in neural networks. James Kirkpatrick, et al., 2017. Proceedings of the National Academy of Sciences.\
> [3] Measuring catastrophic forgetting in neural networks. Ronald Kemker et al., 2017. arXiv.\
> [4] PaLM-E: An Embodied Multimodal Language Model. Danny Driess et al., 2023. arXiv.\
> [5] Lost in the Middle: How Language Models Use Long Contexts. Liu et al., 2023. arXiv.\
> [6] RECKONING: Reasoning through Dynamic Knowledge Encoding. Zeming Chen et al., 2023. arXiv.

---

### Official Review · Reviewer_jgxp · 2023-07-24

**Soundness:** 3

**Excitement:**

3: Ambivalent: It has merits (e.g., it reports state-of-the-art results, the idea is nice), but there are key weaknesses (e.g., it describes incremental work), and it can significantly benefit from another round of revision. However, I won't object to accepting it if my co-reviewers champion it.

**Paper Topic And Main Contributions:**

The paper proposes a new method, Context-aware Meta-learned Loss Scaling (CaMeLS), to dynamically update factual information in online streams. Specifically,  a small auto-regressive meta-network is trained to re-weight the NLL loss for each token. The meta-network is trained by the labeled dataset to help the proxy model adapt factual retention and maintain model behavior at the same time. The experiments show that CaMeLS can learn new knowledge without losing previous abilities.

**Questions For The Authors:**

Question A: Following my reasons to reject, I'd like to see if you have other concerns about other methods and datasets.

Question B: Have you tried other proxy models? (bigger size) I understand there are efficiency problems, but I wonder if bigger proxy models can improve the performance.

**Reasons To Accept:**

1. Unsupervised adaption is an important problem in modern LLMs and this paper's solution has the potential as a general dynamic adapter for deployed static LLMs;

2. The CaMeLS shows strong performance in knowledge retention;

**Reasons To Reject:**

1. The paper lacks comparisons with some baselines, for example, retrieval-augmented, few-shot, and in-context learning methods. When there are some labeled examples, these are other natural solutions;

2. The paper lacks further analysis of the model's generation ability after online adaption. Authors have considered this in Equations (3), but some experiments on other irrelevant datasets (such as SST-2) may help.

**Reproducibility:**

4: Could mostly reproduce the results, but there may be some variation because of sample variance or minor variations in their interpretation of the protocol or method.

**Reviewer Confidence:**

4: Quite sure. I tried to check the important points carefully. It's unlikely, though conceivable, that I missed something that should affect my ratings.

---

> ### Author Rebuttal · Authors · 2023-08-29
>
> Thank you for your review! We're glad you liked the generality of CaMeLS as knowledge updater for pre-trained LLMs.
>
> **Comparisons to other methods:**
> Few-shot prompting is not applicable to our problem setting, where the model must adapt in an *unsupervised* manner to a stream of many (1000+) documents, and is presented with questions about them only at the end of the stream (some supervised data is only assumed present at *meta-training* time). Recent work [1] shows that even models with context lengths long enough to condition on this many documents are not generally able to effectively leverage such long contexts. Additionally, [2] show that gradient-based/parametric model updates can more robustly reason over knowledge compared to in-context learning, which can be more susceptible to distractor facts.
>
> While retrieval is one potential approach to updating LLMs over time, we note that successful use of retrieval is very difficult; recent work [3] finds that LLM-powered search engines only provide a citation for 51.5% of the sentences in LLM-generated responses, and more than 25% of citations do not actually support the associated LLM-generated claim! Further, retrieval adds significant system complexity, making edge deployment far more difficult compared with simple quantized LLM inference that allows large models such as llama-65B to be deployed locally on a user's laptop [4]. For these reasons, we focus on parametric updates to the model's knowledge in this paper.
>
> **Analysis of generation ability:**
> Our current experiments involve unsupervised adaptation of question-answering models. Therefore evaluations of generation quality on other tasks may be uninformative. However, similar unsupervised adaptation of generic generative LMs is an exciting direction for future work!
>
> **Question B**: Please see the answer given to reviewer kJwz.
>
>
>
> [1] Lost in the Middle: How Language Models Use Long Contexts. Liu et al., 2023. arXiv.\
> [2] RECKONING: Reasoning through Dynamic Knowledge Encoding. Zeming Chen et al., 2023. arXiv.\
> [3] Evaluating Verifiability in Generative Search Engines. Liu et al., 2023. arXiv.\
> [4] llama.cpp. Georgi Gerganov, 2023. GitHub. https://github.com/ggerganov/llama.cpp

---

### Official Review · Reviewer_kJwz · 2023-08-12

**Typos Grammar Style And Presentation Improvements:** N/A
**Soundness:** 3

**Excitement:**

4: Strong: This paper deepens the understanding of some phenomenon or lowers the barriers to an existing research direction.

**Missing References:**

None that I could find.

**Paper Topic And Main Contributions:**

This paper proposes a new online method for adapting langauge models to streams of text.

The authors propose a meta-learning approach with three general steps:
    1) a weighting model produces weights over the text
    2) the model is updated with respect to those weights
    3) the weighting model is updated with respect to the adapted model's ability to answer questions about the document

The paper provides ablations of their technique, and analyze its properties (such as which POS tags it upweights) as well as the effect of model on forgetting and answer retention.

**Questions For The Authors:**

* I am a little confused about how, for squad and ArchivalQA, the authors see decreasing performance (ie negative relative F1 improvement) with additional finetuning and uniform weights, especially after QA adaptation.
* In Figure 9 - I think you need to show this effect with more than one document, so you can provide error bars around the line plots.
* Figure 8 - I think it is a stretch to say that your technique best preserves knowledge in the base model, since it leads to worse performance on unrelated questions than uniform weighting.
* Line 498-499: calling the technique "general purpose" by evaluating three datasets (in one task!) is too strong, I would reword this.
* I understand that using Distil-GPT2 for the proxy model makes sense from an efficiency perspective, but are there performance improvements from increasing the size of proxy model?


**Reasons To Accept:**

Overall, while the technique is scoped to QA (with the existence of document-query-label tuples), I think it was a good case study of the technique, and I learned from it. I found this technique to be interesting and new, and quite effective against reasonable baselines (uniform weighting, adaptation for QA, and "salient spans"). I thought the authors had careful ablations that showed their technique was not just putting higher weight on nouns/pronouns.

While the authors note that binarizing the weights does not produce substantially different outcomes, I am not sure if that is necessarily a weakness; it might just indicate that the relative weighting of different tokens may not matter as much, which simplifies the approach slightly.

The authors test their approach up to 6B parameters, showing that their technique brings consistent improvements across model scales.



**Reasons To Reject:**

* I don't see mention of code release, which would be helpful for reproducibility.
* I think a simple baseline, using TF-IDF weighting (perhaps with some thresholding to, for example, remove stopwords and extremely common tokens), might be a much simpler and cheaper alternative to the proposed technique. It would be nice to see this comparison in the paper.

**Reproducibility:**

2: Would be hard pressed to reproduce the results. The contribution depends on data that are simply not available outside the author's institution or consortium; not enough details are provided.

**Reviewer Confidence:**

3: Pretty sure, but there's a chance I missed something. Although I have a good feel for this area in general, I did not carefully check the paper's details, e.g., the math, experimental design, or novelty.

---

> ### Author Rebuttal · Authors · 2023-08-29
>
> Thank you for the review. We are glad you find CaMeLS to be an interesting new approach!
> >I don't see mention of code release, which would be helpful for reproducibility.
>
> We unfortunately forgot to mention this. There will be a public code release. Until then, we have also anonymized the code which can be found here https://anonymous.4open.science/r/CaMeLS-63F8/.
>
> >I think a simple baseline, using TF-IDF weighting (perhaps with some thresholding to, for example, remove stopwords and extremely common tokens), might be a much simpler and cheaper alternative to the proposed technique. It would be nice to see this comparison in the paper.
>
> Thank you for the suggestion. We have preliminary results showing TF-IDFs performance adapting GPT2-XL and gpt-neo-1.3B on streamingQA. We additionally show the standard error of mean over 4 seeds. TF-IDF Cutoff additionally removes the 5% of words with lowest TFIDF. We find TF-IDF to be an effective baseline for online adaptation, performing slightly better than salient span masking. **However, CaMeLS still enables significantly better knowledge uptake than TF-IDF.** We will add an extensive comparison to the final paper.
>
> | Method        | GPT-Neo-1.3B F1 Increase | GPT2-XL F1 Increase  |
> |---------------|--------------------------|----------------------|
> | CaMeLS        |  0.190 ± 0.017           | 0.308 ± 0.018        |
> | TF-IDF        |  0.073 ± 0.006           | 0.034 ± 0.008        |
> | TF-IDF Cutoff |  0.074 ± 0.005           | 0.082 ± 0.020        |
> | Uniform       | -0.099 ± 0.014           | 0.038 ± 0.026        |
> | Uniform + QA  | -0.016 ± 0.035           | 0.178 ± 0.041        |
> | SSM           |  0.058 ± 0.018           | 0.036 ± 0.041        |
>
>
> >I am a little confused about how, for squad and ArchivalQA, the authors see decreasing performance (ie negative relative F1 improvement) with additional finetuning and uniform weights, especially after QA adaptation.
>
> We agree that negative relative F1 improvements are unintuitive.  This decrease in performance is primarily observed in the GPT-Neo 1.3B model, which is among the weaker performers across all datasets and adaptation methods examined. In these particular settings where F1 scores decline, we still observe a decrease in the Negative Log-Likelihood (NLL) for answer tokens following the questions. This suggests that the model is still adapting to the new text to some extent. We hypothesize that while the models do gain new information about the documents during the adaptation phase, they become less adept at responding to question prompts in the desired format. (even after additional QA tuning). In continual learning literature, work has found that after training on many different data distributions, the ability to learn the next data distribution diminishes. aka, neural nets lose plasticity over time [1]. Even for models which show an increase in F1, we see that QA-tuning post adaptation does not fully recover initial model performance on unrelated question-answer pairs (Figure 8, right).
> >Figure 8 - I think it is a stretch to say that your technique best preserves knowledge in the base model, since it leads to worse performance on unrelated questions than uniform weighting.
>
> >Line 498-499: calling the technique "general purpose" by evaluating three datasets (in one task!) is too strong, I would reword this.
>
> Thank you for the critiques of particular figures and claims. We will reword the specified points to be more accurate. Figure 9 currently shows document performance averaged over all documents presented over multiple online adaptation experiments, not performance for just a single fact. We apologize for any confusion and will update the figure to be more clear and to show error bars.
>
> >I understand that using Distil-GPT2 for the proxy model makes sense from an efficiency perspective, but are there performance improvements from increasing the size of proxy model?
>
> We are also curious about the effect of using larger proxy models. We trained a CaMeLS weight model using GPT2-small (114M params) as a proxy model. Due to compute limitations we were not able to meta-train using any larger proxy models. Results comparing performance between the two proxy model sizes on StreamingQA are shown below, errors are 1 standard error of mean.
> | Proxy Model       | GPT-Neo-1.3B F1 Increase | GPT2-XL F1 Increase |
> |-------------------|-------------------------|---------------------|
> | DistilGPT2 (82M)  | 0.190 ± 0.017           | 0.308 ± 0.018       |
> | GPT2-Small (117M) | 0.176 ± 0.023           | 0.309 ± 0.012       |
>
> We see no significant difference in performance in this setting. Qualitatively, the two weighting models generate similar outputs. We hypothesize that CaMeLS learns a weighting which reflects the innate importance of tokens in the text to answering the meta-training questions, rather than a proxy model specific token importance. If this were the case, the performance of CaMeLS would not always depend on the proxy model used. We emphasize that this is a hypothesis and believe a more rigorous exploration of proxy model size is an exciting direction for future work.
>
> [1] Continual Backprop: Stochastic Gradient Descent with Persistent Randomness. Shibhansh Dohare, et al., 2021. arXiv.

---

### Official Review · Reviewer_dDnG · 2023-08-12

**Soundness:** 4

**Excitement:**

4: Strong: This paper deepens the understanding of some phenomenon or lowers the barriers to an existing research direction.

**Paper Topic And Main Contributions:**

The paper focuses on improving unsupervised online language model adaptation by proposing a meta-learning algorithm (CaMeLS) that learns an importance weighing model to re-weight the per-token loss of the data-stream. The main hypothesis of the paper is that online adaptation by naive fine-tuning can be improved by only fine-tuning on a subset of tokens in the data-stream, which likely lead to useful updates.  CaMeLS makes use of a small proxy model like DistilGPT-2 to train the weighting model, which uses an episodic bi-level optimization for training. CaMeLS also uses an additional corpus of documents having query samples and corresponding labels to identify which tokens in the documents are most likely to be informative. The proposed approach was evaluated on 3 datasets:  StreamingQA, SQuAD and ArchivalQA.


**Questions For The Authors:**

a.  Could you please provide more details on how the additional labeled dataset was generated to train the weighing model?

b. What is added computational overhead of CaMeLS when compared to the baseline? Please consider including this comparison in the paper

**Reasons To Accept:**

- The paper proposes a novel approach that improves knowledge update after online language model adaption on a data stream. CaMeLS improves F1 score and  outperforms baselines like naive fine-tuning across different datasets and models.
- The paper has ablation experiments to understand the importance of context aware weights for knowledge retention. These insights might be valuable to the community
- Experiments also demonstrate that the CaMeLS approach can be generalizable on unseen data distributions

**Reasons To Reject:**

-  CaMeLS assumes an additional corpus of labeled data to train the weighting model. However, it is unclear how this dataset was generated. This assumption will be a problem when the labeling process is expensive.
- The paper lacks comparisons on the compute overhead required for CaMeLS compared to the baselines
- Writing needs to be improved. Many sections are hard to follow

**Reproducibility:**

3: Could reproduce the results with some difficulty. The settings of parameters are underspecified or subjectively determined; the training/evaluation data are not widely available.

**Reviewer Confidence:**

3: Pretty sure, but there's a chance I missed something. Although I have a good feel for this area in general, I did not carefully check the paper's details, e.g., the math, experimental design, or novelty.

---

> ### Author Rebuttal · Authors · 2023-08-29
>
> Thank you for the review and the thoughtful feedback! We're glad you appreciated the novelty of CaMeLS and found our experiments insightful.
>
>  >CaMeLS assumes an additional corpus of labeled data to train the weighting model. However, it is unclear how this dataset was generated. This assumption will be a problem when the labeling process is expensive.
>
> It is true that CaMeLS requires a corpus of document query pairs to train the weighting model. Many question-answering datasets - including the three we study - are generated by first selecting documents and then creating corresponding questions for the documents (via LLMs or humans) [1,2,3]. Our experiments with CaMeLS use the StreamingQA, SQuAD, and ArchivalQA data sets off the shelf. The only additional step needed was matching the questions with their original source documents - that is, given a document ID, retrieve the original document from the source corpus. Our dependence on labeled data is further mitigated by the ability of CaMeLS to generalize to unseen data distributions (4.6). We are excited about learning CaMeLS weight models using only unlabeled data as a future direction of research.
>
>  >The paper lacks comparisons on the compute overhead required for CaMeLS compared to the baselines
>
> We discuss the compute costs of training CaMeLS in Appendix B. In the comparison below, we find that **the additional compute needed for CaMeLS is very small compared to uniform fine-tuning.** When using CaMeLS for online adaptation, the compute overhead of CaMeLS is a single forward pass of a weight model for each document we update on. For large models, the weight model overhead is small compared to the time needed to run a forward pass and backprop of the base model. Performances of the two methods when used to adapt GPT2-XL on StreamingQA documents are shown in the table below. Performances are logged on a NVIDIA A100 (80GB, SXM4). CaMeLS requires slightly more GPU memory to store the weight model, and is marginally slower per document.
>
> | Model         | Time per Document (± std) | GPU Memory Allocated |
> |---------------|-------------------------------|----------------------|
> | GPT2-XL, Uniform | 772.72 ± 25.95 ms         | 46.62 GB              |
> | GPT2-XL, CaMeLS  | 782.46 ± 26.86 ms         | 48.18 GB              |
>
>
>
>
> >Writing needs to be improved. Many sections are hard to follow
>
> Apologies for the unclear sections. We will further edit and improve the paper. Feel free to let us know which sections were the hardest to follow.
>
> [1] StreamingQA: A Benchmark for Adaptation to New Knowledge over Time in Question Answering Models. Adam Liška, et al., 2022. arXiv.\
> [2] ArchivalQA: A Large-scale Benchmark Dataset for Open Domain Question Answering over Historical News Collections. Jiexin Wang, et al., 2021. arXiv.\
> [3] SQuAD: 100,000+ questions for machine comprehension of text. Pranav Rajpurkar, et al., 2016. arXiv.

---

### Meta-Review · Area_Chair_uAXm · 2023-09-19

**Recommendation:** 5

**Metareview:**

This paper proposes learning which tokens to upweight in the training of language models, in order to adapt the model to new sources of information. They propose Context-aware Meta-learned Loss Scaling (CaMeLS), which uses an auxiliary language model to learn how to reweight the language modeling loss for each token.

Reviewers recognize the novelty of this work, and find the method well motivated. Some raise questions about the efficiency of the approach which the author response addresses. More interestingly, a suggested baseline by a reviewer is implemented by the authors and compared to their own results, which still highlights the effectiveness of CaMeLs. Some proposed baselines by the reviewers are also compared with in the author response. Overall the reviewers seem satisfied with the author response.

We hope the authors include the new results and suggested changes in the next iteration of the paper.

---

### Decision · Program_Chairs · 2023-10-07

**Decision:**

Accept-Main

**Comment:**

This paper proposes learning which tokens to upweight in the training of language models, in order to adapt the model to new sources of information. They propose Context-aware Meta-learned Loss Scaling (CaMeLS), which uses an auxiliary language model to learn how to reweight the language modeling loss for each token.

Reviewers recognize the novelty of this work, and find the method well motivated. Some raise questions about the efficiency of the approach which the author response addresses. More interestingly, a suggested baseline by a reviewer is implemented by the authors and compared to their own results, which still highlights the effectiveness of CaMeLs. Some proposed baselines by the reviewers are also compared with in the author response. Overall the reviewers seem satisfied with the author response.

We hope the authors include the new results and suggested changes in the next iteration of the paper.